green chemistry/nanotechnology/organic chemistry

green chemistry, bionanocomposite, starch, CuFe$_2$O$_4$, 4H-pyran, nanocatalyst

**Author for correspondence:**
Ali Maleki
e-mail: maleki@iust.ac.ir

This article has been edited by the Royal Society of Chemistry, including the commissioning, peer review process and editorial aspects up to the point of acceptance.

# Green and efficient three-component synthesis of 4H-pyran catalysed by CuFe$_2$O$_4$@starch as a magnetically recyclable bionanocatalyst

## Maryam Kamalzare[1], Mohammad Bayat[1] and Ali Maleki[2]

[1]Department of Chemistry, Faculty of Science, Imam Khomeini International University, Qazvin, Iran
[2]Catalysts and Organic Synthesis Research Laboratory, Department of Chemistry, Iran University of Science and Technology, Tehran 16846-13114, Iran

MK, 0000-0003-0029-5534; MB, 0000-0002-5235-1203; AM, 0000-0001-5490-3350

The development of simple, practical and inexpensive catalysis systems using natural materials is one of the main goals of pharmaceutical chemistry as well as green chemistry. Owing to the ability of easy separation of nanocatalyst, those goals could be approached by applying heterogeneous bionanocatalyst in combination with magnetic nanoparticles. Starch is one of the most abundant natural polymers; therefore, preparing bionanocatalyst from starch is very valuable as starch is largely available and inexpensive. An ecologically benign and efficacious heterogeneous nanocatalyst was prepared based on a biopolymer, and its attributes and morphology were specified by using Fourier transform infrared spectra, scanning electron microscopy (SEM), energy-dispersive X-ray spectroscopy (EDX), X-ray diffraction (XRD), thermal analysis and vibrating sample magnetometer measurements; followed by studying catalytic behaviour of bionanocomposite in a multicomponent reaction to synthesize of 4H-pyran derivatives. 4H-pyran is extremely valuable in pharmaceutical chemistry, and the development of methods for synthesis of different derivatives of 4H-pyran is momentous. Revealing environmentally benign nature, mild condition, easy work-up, low cost and non-toxicity are some of the advantages of this protocol. Besides, the bionanocomposite was recovered using an external magnetic bar and could be re-used at least six times with no further decrease in its catalytic activity.

# 1. Introduction

Developing a new generation of hybrid nanostructured materials, the green nanomaterial is currently a primary objective of research in nanotechnology. The recent study on bionanocomposite has turned into an emerging field for chemists due to the quality of bionanocomposites in eliminating or minimizing wastes and implementing sustainable processes. That is, a combination of biopolymers and inorganic nanoparticles leads to the development of new novel materials with distinctive properties to be used in various applications [1,2]. The specifically large surface area of nanoparticles makes them extremely active; thus, green chemistry has focused on preparing the bionanocomposite using a biopolymer to serve as a protecting agent for nanoparticles [3]. Hence, choosing polymers from renewable resources has recently received great attention from chemists, and using biodegradable and renewable polymers for various applications has become highly desirable [4]. Starch is one of the most favourite sources of biopolymers. Owing to its valuable characteristics such as low cost, biodegradability and extensive accessibility, starch has gained much attention [5].

On the other hand, magnetic nanoparticles are widely used as parts of nanomaterials and, therefore, are being used in various applications such as serving as catalysts as well as in medical sciences. One of the most important challenges in the catalyst industry is the easy dissociation of nanocatalyst from the reaction media. The superparamagnetic nature of magnetic material causes its detachment from the reaction mixture by an external magnetic bar. This unique property removes the necessity of catalyst purification as well as a tedious and hard work-up upon reaction termination and prevents the loss of catalyst [6,7].

The challenging task of achieving a simple, cost-effective and green reaction procedure for medicinal chemistry is a noteworthy area of research in both academic and pharmaceutical studies. Multicomponent reactions (MCRs), which can be one of the most beneficial protocols in synthetic chemistry, start from simple materials [8,9]. On the other hand, fused benzo 4H-pyran, to be specific, the 4H-chromene derivatives, have been recognized as an important class of natural oxygen-containing heterocyclic compounds that are extensively available in fruits and vegetables. 4H-pyran offers different biological and pharmaceutical activities such as antifungal, antiviral, antioxidant, antileishmanial, antiallergenic, antibacterial, hypotensive, anticoagulant, diuretic and anti-tumor activities [10,11] (figure 1). Therefore, proceeding with the synthesis of such complex heterocyclic compounds has been considered as one of the most interesting fields by both organic and medicinal chemists. Various ways have been reported through modifying the synthesis of 4H-pyran and its derivatives using diverse catalysts and different reaction conditions. Several improved reactions have been presented which use a variety of catalysts such as PEG1000-DAIL [12], hexadecyltrimethylammonium bromide (HTMAB) [13], sodium selenate [14], tetra-methyl ammonium hydroxide [15], amberlite IRA400 (OH-) [16] and MgO [17]. Some of the previous methods have posed a number of problems such as lengthy procedures, being highly corrosive, using expensive catalysts, volatile solvent, laborious work-up and high energy consumption caused by the high-temperature reaction. Furthermore, there has been an intense demand for developing a new and green protocol for synthesizing 4H-pyran derivatives. There are several heterogeneous catalysts for the synthesis of 4H-pyran derivatives. Highly efficient triazine functionalized ordered mesoporous organosilica as a unique metal-free organocatalyst under solvent-free conditions for the three-component reaction in synthesizing 4H-pyran derivatives. Easy work-up, being waste-free, and the ability to re-use catalysts are among the advantages of this method [18]. Tungstic acid functionalized mesoporous SBA-15 is another efficient heterogeneous catalyst for one-pot synthesis of 4H-pyran in water as a solvent. This catalyst system has a high surface area and good acidity; thus, it can catalyse the reaction efficiently and with high output. Furthermore, this catalyst can be recycled multiple times without further decrease in its catalytic activity [19].

In this regard, in order to continue our previous research on magnetically recyclable nanocatalysts in MCRs [20–24], we hereby propose $CuFe_2O_4$@starch bionanocatalyst for the three-component reaction of the aromatic aldehyde, malononitrile and enolizable C-H activated acidic in ethanol as a solvent for the synthesis of 4H-pyran derivatives at room temperature (scheme 1).

One of the arguments in favour of the novelty of $CuFe_2O_4$@starch, in comparison with another heterogeneous catalyst, is bionanocomposite synthesized from biocompatible materials makes the synthesis of 4H-pyran derivatives more eco-friendly. In addition, due to the $CuFe_2O_4$ nanoparticles, the bionanocatalyst has a larger active surface area that generates higher activity as well as a high yield of pure product. Besides, it can be recycled and re-used frequently, and the work-up procedure and separation of catalyst from reaction media are straightforward, as it requires only an external

**Figure 1.** Selected examples of pyran derivatives with pharmaceutical and biological activity.

**Scheme 1.** One-pot three-component reaction of different enolizable C-H activated acidic compounds, aldehydes and malononitrile catalysed by CuFe$_2$O$_4$@starch in ethanol at room temperature.

magnet. CuFe$_2$O$_4$@starch is stable, resistant and is economically affordable owing to its simple synthesis procedure from commercially available materials, generating the small amounts of chemical wastes, having short reaction times and going through easy separation steps.

# 2. Material and methods

## 2.1. Chemicals and instruments

The solvents, chemicals and reagents were purchased from Merck, Fluka and Aldrich chemical companies and were used without further purification. The melting points were measured on an Electrothermal 9100 apparatus. The IR spectra were recorded on a Shimadzu IR-470 spectrometer in KBr pellets procedures. Scanning electron microscopy (SEM) images were taken with VEGA2 TESCAN, the statistical data of particle sizes from FE-SEM imaging was obtained by Digimizer software. The X-ray (EDX) analysis was recorded with a Numerix DXP–X10P. Thermal analysis was performed by Bahr-STA 504 instrument in the air atmosphere. XRD patterns of the solid powders were developed using a JEOL JDX–8030 (30 kV, 20 mA), the NMR spectra were recorded by Bruker DRX-300 Advance instrument (300 MHz for HNMR and 75.4 MHz for CNMR) while DMSO was used as a solvent. The chemical shifts are given in parts per million (ppm), and the coupling constants (*J*) are reported in hertz (Hz) scales. Merck starch gel GF254 plates were used for the analytical thin-layer chromatography (TLC) procedure.

## 2.2. Synthesis of CuFe$_2$O$_4$ nanoparticles

CuFe$_2$O$_4$ nanoparticles were prepared through thermal decomposing of Cu(NO$_3$)$_2$ and Fe(NO$_3$)$_3$ in water in the presence of sodium hydroxide. To describe it briefly, first Fe(NO$_3$)$_3$·9H$_2$O (3.3 g, 8.2 mmol) and Cu(NO$_3$)$_2$·3H$_2$O (1 g, 4.1 mmol) were dissolved in 75 ml of distilled water, then 3 g (75 mmol) of NaOH was dissolved in 15 ml of distilled water and added to it at room temperature over a period of 10 min, during which a reddish-black precipitate was formed. Next, the reaction mixture was put in an ultrasound device at 90°C. After 2 h, it was cooled at room temperature. An external magnetic bar was then used to separate the synthesized magnetic nanoparticles. The procedure was followed by washing the particles with distilled water several times and drying it in an air oven overnight at 80°C. Finally, the nanoparticles were ground in a mortar and pestle and were kept in a furnace at 700°C for 5 h (step-up temperature 20°C per minute). The product was then cooled at room temperature slowly. The result was procuring 820 mg of magnetic CuFe$_2$O$_4$ nanoparticles.

## 2.3. Synthesis of CuFe$_2$O$_4$@starch nanocomposite

About 0.1 g CuFe$_2$O$_4$ nanoparticles were dispersed in 5 ml distilled water by ultrasonic waves; parallel to that, 0.9 g starch was dissolved in 10 ml distilled water; these two were mixed and were stirred at room temperature for 8 h. The resulting solution was cast and dried on a glass plate for 48 h to afford starch-supported magnetic nanocomposite.

## 2.4. General procedure for the synthesis of 4H-pyran derivatives

The mixture of aryl aldehyde (1 mmol), enolizable C-H activated acidic compounds (**3a, 3b, 5, 7**) (1 mmol) and malononitrile (1.1 mmol) with the presence of CuFe$_2$O$_4$@starch (0.03 g) as a catalyst in 3 ml of ethanol as a solvent of the reaction stirred for an appropriate time at room temperature. The completion of the reaction was monitored by TLC (ethyl acetate/$n$-hexane, 1 : 3). The magnetic nanocomposite was easily removed with the help of a magnetic stirring bar and external magnetic bar as soon as the stirring was stopped, followed by filtering the reaction solution. The precipitate was then purified by washing with ethanol. The pure products were obtained from the reaction media. The CuFe$_2$O$_4$@starch was then cleaned up with EtOH, was air-dried and used for other reactions without a significant loss of its catalytic characteristics after frequent usage.

## 2.5. Spectral data for selected products

*2-amino-7,7-dimethyl-4-(4-nitrophenyl)-5-oxo-5,6,7,8-tetrahydro-4H-chromene-3-carbonitrile (4b):* White solid: m.p. 180–182°C, yield: 94%. $^1$H NMR (300 MHz, DMSO-$d6$): $\delta$ = 0.93 (3H, s, CH$_3$), 1.02 (3H, s, CH$_3$), 2.06 (2H, dd, CH$_2$, $^3J$ = 15 Hz), 2.22 (2H, dd, CH$_2$, $^3J$ = 15 Hz), 4.34 (1H, s, CH), 7.18 (2H, br s, NH$_2$), 7.41 (2H, d, Ar, $^3J$ = 8.1 Hz), 8.14 (2H, d, Ar, $^3J$ = 5.4 Hz). $^{13}$C NMR (75 MHz, DMSO-$d6$): 27.37 (2CH$_3$), 28.78 (C(CH$_3$)), 32.07 (CH$_2$), 35.95, 50.3 (CH$_2$), 57.99 (C(CN)), 112.09, 119.89, 124.30 (CN), 129.30, 146.7, 152.81, 159.16, 163.55, 196.

*2-amino-7,7-dimethyl-4-(2-nitrophenyl)-5-oxo-5,6,7,8-tetrahydro-4H-chromene-3-carbonitrile (4c):* White solid: m.p.: 224–226°C, yield: 91%. $^1$H NMR (300 MHz, DMSO-$d6$): $\delta$ = 0.86 (3H, s, CH$_3$), 0.99 (3H, s, CH$_3$), 1.96 (2H, dd, CH$_2$, $^2J$ = 15.9 Hz), 2.16 (2H, dd, CH$_2$, $^2J$ = 15 Hz), 4.91 (1H, s, CH), 7.19 (2H, br s, NH$_2$), 7.32 (1H, d, Ar, $^3J$ = 7.5 Hz), 7.32–7.67 (1H, m, Ar), 7.78 (1H, d, Ar, $^3J$ = 8.1 Hz). $^{13}$C NMR (75 MHz, DMSO-$d6$): $\delta$ = 27.21 (2CH$_3$), 28.75 (C(CH$_3$)), 30.72(CH), 32.31 (CH$_2$), 50.34 (CH$_2$), 56.72 (C(CN)), 113.1, 119.48, 124.24 (CN), 128.39, 130.59, 134.02, 139.42, 149.59, 159.74, 163.22, 196.48.

# 3. Results and discussion

## 3.1. Characterization of CuFe$_2$O$_4$@starch nanocomposite

The nanocomposite containing CuFe$_2$O$_4$ nanoparticles was prepared and characterized by various analyses. The comparative IR spectra of CuFe$_2$O$_4$@starch and starch are shown in figure 2. The peaks in the 3200–3800 cm$^{-1}$ range were corresponding to the OH stretching vibrations in the starch. The broadening in this region suggested the intermolecular hydrogen bonding, and the C-H stretching was observed at the wavelength of 2923 cm$^{-1}$. The peak appearing at 1643 cm$^{-1}$ is related to OH bending vibration, and the peaks at 1157, 1083 and 1000 cm$^{-1}$ are related to C-O bond stretching. In the FT-IR spectra of CuFe$_2$O$_4$@starch, hydroxyl group signals, which are corresponded to starch, have been decreased. This happens because of the chelation of copper and iron atoms with the oxygen atoms of starch. The characteristic bands of Cu-O and Fe-O at 474 and 621 cm$^{-1}$, respectively, represented the metal-oxygen group of CuFe$_2$O$_4$. The CuFe$_2$O$_4$@starch was characterized by SEM analysis to find its morphology and size (figure 3$a$). According to the SEM image, the average size of synthesizing CuFe$_2$O$_4$ nanoparticles is less than 50 nm, and CuFe$_2$O$_4$ nanoparticles were loaded on the surfaces of the starch. There was a potent interaction between the nanoparticles and the starch matrix, due to which, the dispersed CuFe$_2$O$_4$ nanoparticles in the composite were considered as sustained. To specify the size of nanoparticles, 50 particles were used at random. The mean particle size of nanoparticles, as analysed by Digimizer software, is about 47 nm (figure 3$b$).

The XRD pattern was prepared and analysed to specify the structure of the inorganic nanoparticles in the bionanocomposite. As shown in figure 4, the nanocomposite exhibited the main peaks, which were consistent with the characteristic peaks of CuFe$_2$O$_4$ (JCPDS No. 77-0010), indicating the peaks at the

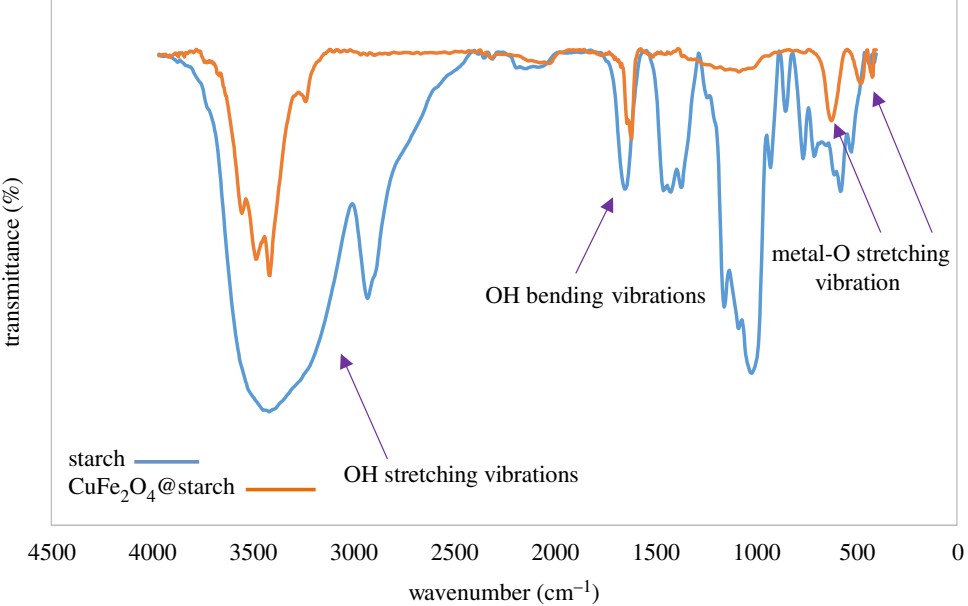

**Figure 2.** IR spectra of starch and CuFe$_2$O$_4$@starch bionanocomposite.

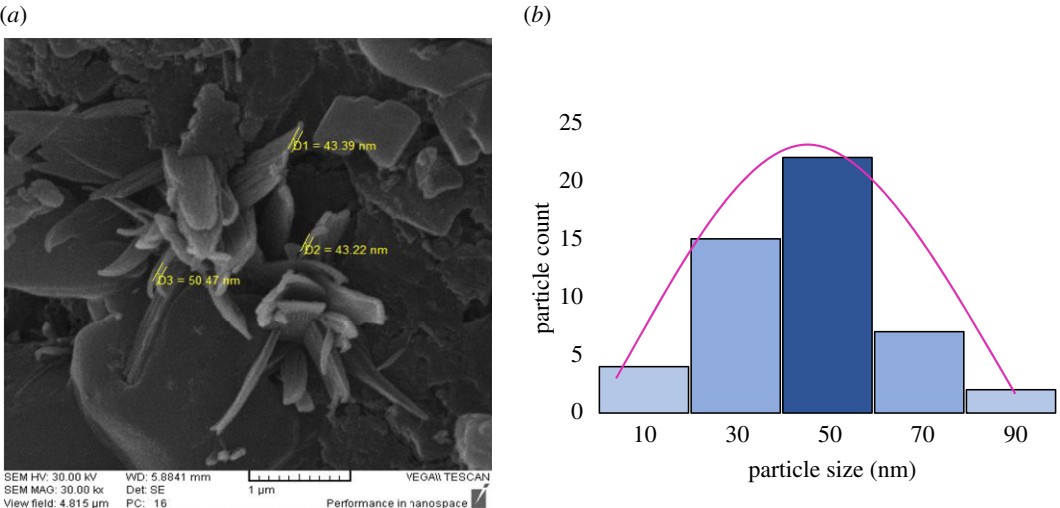

**Figure 3.** (*a*) SEM image of CuFe$_2$O$_4$@starch bionanocomposite and (*b*) the particle size distribution diagram of the CuFe$_2$O$_4$@starch calculated by Digimizer software.

dispersion angle $2\theta = 30.2°$, $35.4°$, $37.1°$, $42.9°$, $53.4°$, $56.8°$, $62.4°$ and $73.7°$ are the main peaks of CuFe$_2$O$_4$. In addition, the mean size of the nanoparticles characterized by X-ray line broadening using the Scherrer equation ($D = k\lambda/\beta \cos\theta$) was about 43 nm.

The magnetic property of CuFe$_2$O$_4$@starch bionanocomposite was studied by a vibrating sample magnetometer (VSM) at room temperature (figure 5). The VSM magnetization curve of CuFe$_2$O$_4$@starch proved that the magnetic property of bionanocomposite was sufficient to detach easily from the reaction media using an external magnet. The TGA analysis has shown the thermal stability of bionanocomposite. The procedure was carried out in the air atmosphere and underwent heating over the temperature range of 50–1000°C. CuFe$_2$O$_4$@starch showed desirable thermal stability. The minor increase, as seen at the beginning of the graph, is caused by adsorbing of the surrounding moisture. The loss of weight in around 100–200°C temperature is due to the removal of the surface adsorbed water and other solvents, or, because of the molecules physically adsorbed onto the surface of the bionanocomposite. The reduction in weight, as observed in around 300–400°C, is due to the organic structure of bionanocomposite. The graph shows that after 400°C, there will be no significant change in the bionanocomposite mass, being an indication of the presence of nanoparticles in the

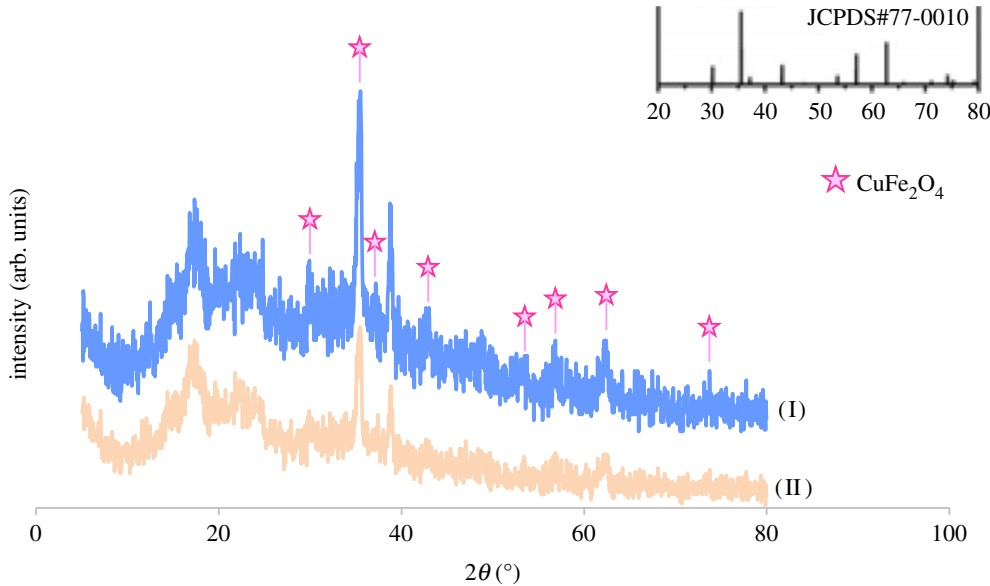

**Figure 4.** The XRD pattern of (I) CuFe₂O₄@starch bionanocomposite and (II) re-used bionanocomposite.

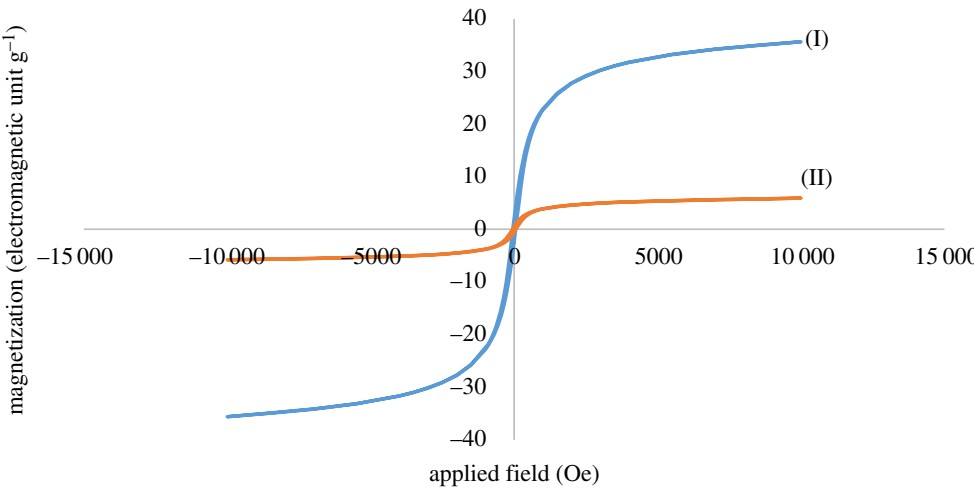

**Figure 5.** VSM magnetization curve of (I) CuFe₂O₄ (II) CuFe₂O₄@starch bionanocomposite.

bionanocomposite (figure 6). The fact that up to 300°C there is no considerable reduction in the mass of CuFe₂O₄@starch indicates the synthesized bionanocomposite has good thermal stability in multicomponent reaction procedures.

The EDX analysis shows the chemical mixture of bionanocomposite. This analysis indicates the engagement of Fe, Cu, C and O atoms elements in the present bionanocomposite (figure 7a); hence, proving CuFe₂O₄ nanoparticles are loaded on the surface of starch. As both organic and the inorganic parts of the prepared bionanocatalyst contain an oxygen atom, the highest percentage of weight is related to the oxygen element. Furthermore, the elemental mapping of EDX patterns shows the presence of Fe, Cu, O and C elements in the bionanocomposite (figure 7b).

## 3.2. The catalytic activity of CuFe₂O₄@starch bionanocomposite as nanocatalyst

### 3.2.1. Application CuFe₂O₄@starch for the synthesis of 4H-pyran derivatives

To demonstrate the catalytic activity of the synthesized bionanocomposite, we studied the application of CuFe₂O₄@starch as heterogeneous bionanocatalyst in the synthesis of 4H-pyran derivatives.

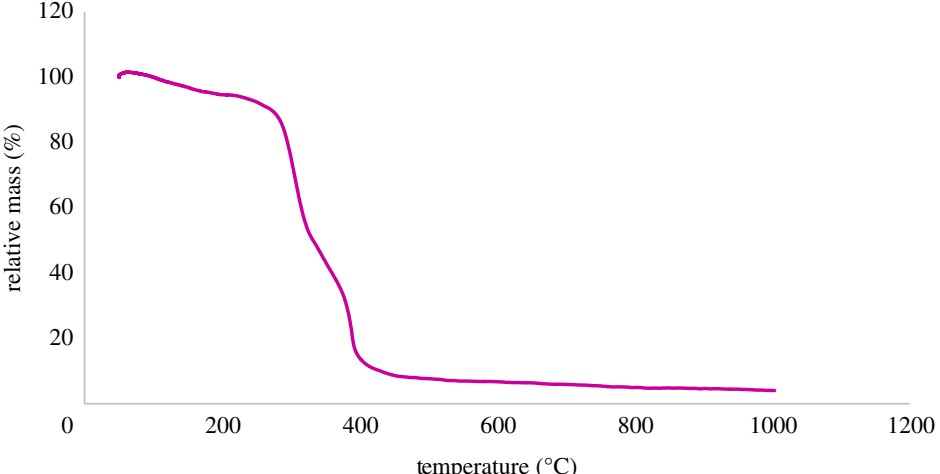

**Figure 6.** TGA curves of CuFe$_2$O$_4$@starch bionanocomposite.

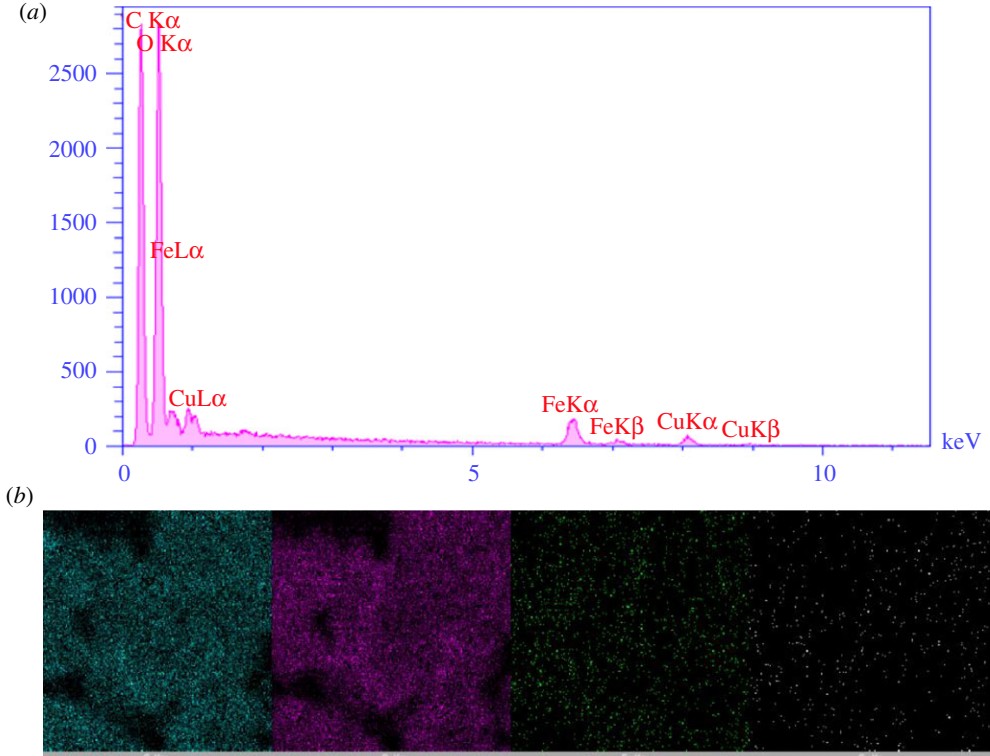

**Figure 7.** (*a*) EDX analysis of CuFe$_2$O$_4$@starch bionanocomposite and (*b*) elemental mapping of CuFe$_2$O$_4$@starch bionanocomposite.

The three-component reaction of 4-chlorobenzaldehyde, malononitrile and dimedone was chosen as the model reaction to optimize the reaction conditions. The outcome of applying different solvents, temperature and various amounts of CuFe$_2$O$_4$@starch bionanocatalyst on the reaction efficiency was studied. As it can be seen in table 1, the reaction efficiency in the absence of any catalyst, with water or ethanol at room temperature or reflux condition, was traced. The outcome of the product in H$_2$O as a green solvent was significantly less than EtOH under similar reaction conditions. According to the observed results, the optimum amount of bionanocatalyst was 30 mg. To study the generality of this method, three different reaction categories were applied for the synthesis of 4H-pyran derivatives after optimizing the reaction conditions.

To prove and expand the extent of this new effective methodology, the optimized reaction conditions were developed to cover other enolizable C-H activated acidic compounds, including dimedone and ethyl acetoacetate. As table 2 shows, aromatic aldehydes with electron-withdrawing groups react

**Table 1.** Optimization of the reaction conditions for preparation of 2-amino-4-(4-chlorophenyl)-7,7-dimethyl-5-oxo-5,6,7,8-tetrahydro-4H-chromene-3-carbonitrile.

| entry | solvent | temp. (°C) | catalyst (mg) | yield[a] (%) |
|---|---|---|---|---|
| 1 | EtOH | room temperature | — | Trace |
| 2 | EtOH | reflux | — | Trace |
| 3 | $H_2O$ | room temperature | — | Trace |
| 4 | $H_2O$ | reflux | — | Trace |
| 5 | EtOH | room temperature | 20 | 65 |
| 6 | EtOH | room temperature | 30 | 96[b] |
| 7 | EtOH | room temperature | 40 | 96 |
| 8 | EtOH | reflux | 30 | 90 |
| 9 | $H_2O$ | room temperature | 30 | trace |
| 10 | $CH_3CN$ | room temperature | 30 | 89 |
| 11 | $CH_3CN$ | reflux | 30 | 91 |

[a]Isolated yield for product **4a**, *via* coupling reaction between malononitrile (1.1 mmol) and 4-chlorobenzaldehyde (1.0 mmol), dimedone (1.0 mmol), in the presence of $CuFe_2O_4$@starch (30 mg) as a catalyst and ethanol as a solvent at room temperature.
[b]Optimum conditions.

faster than aromatic aldehydes with electron-donating groups. Furthermore, the three-component reaction with ethyl acetoacetate (**3b**), as an acyclic 1,3-dicarbonyl, required a longer reaction period compared to dimedone (**3a**) under similar reaction conditions (scheme 2).

To reveal the potency of bionanocatalyst, the optimized reaction conditions were developed to include another enolizable C-H activated acidic compound, 4-hydroxycoumarin, on the same reaction conditions (table 3 and scheme 3).

Finally, different derivatives of 4H-chromene were synthesized from the three-component reaction of malononitrile, various aldehyde and 2-hydroxy-1,4-naphthoquinone to serve as the enolizable C-H activated acidic compound in $CuFe_2O_4$@starch as a heterogeneous bionanocatalyst (table 4 and scheme 4).

## 3.3. Catalyst recycling

The ability to recover and re-use bionanocatalyst as a heterogeneous catalyst is one of the most critical properties of bionanocomposite that makes it valuable, unique and beneficial. To sum up, after completion of the reaction, bionanocatalyst was magnetically separated from the reaction media. It was washed with ethanol and was dried at room temperature. The reusability of the bionanocomposite was studied on the model reaction as well. $CuFe_2O_4$@starch can be recycled and re-used six times with 96, 93, 93, 92, 90 and 89% of product yields, respectively (figure 8).

## 3.4. Reaction mechanism

A proposed mechanism for the synthesis of 4H-pyran derivatives catalysed by $CuFe_2O_4$@starch is shown in figure 9. As it can be seen, $CuFe_2O_4$@starch acted as Lewis acid and was also able to increase the electrophilicity of carbonyl groups. First, aldehyde and dimedon were condensed together through the Knoevenagel condensation process. Then, this intermediate reacted as a Michael acceptor; thus, the attack of enolizable C-H activated acidic compounds to this molecule led to an open-chain intermediate. Finally, an intramolecular cyclization of this intermediate gave the desired products.

## 4. Conclusion

In this study, we have synthesized green heterogeneous bionanocatalyst from inexpensive, readily available and natural materials with distinctive properties such as being eco-friendly, inexpensive and showing high efficiency for the synthesis of 4H-pyran derivatives. In terms of efficiency and eco-

**Table 2.** Three-component synthesis of different 2-amino-5-oxo-5,6,7,8-tetrahydro-4H-benzo[b]pyrans (**4a–u**) via condensation of malononitrile (**1**), various aldehydes (**2**) and dimedone (**3a**) or ethyl acetoacetate (**3b**) in the presence of $CuFe_2O_4$@starch at room temperature.

| entry | aldehyde | product | time (min) | isolated yields (%) | m.p. (obsd) (°C) | m.p. (lit) (°C) |
| --- | --- | --- | --- | --- | --- | --- |
| 1 | 4-chlorobenzaldehyde | **4a** | 20 | 96 | 246 | 240–242 [25] |
| 2 | 4-nitrobenzaldehyde | **4b** | 25 | 94 | 180–182 | 180–182 [11] |
| 3 | 2-nitrobenzaldehyde | **4c** | 25 | 91 | 224–226 | 223–225 [26] |
| 4 | 3-bromobenzaldehyde | **4d** | 30 | 92 | 211–215 | 224–226 [27] |
| 5 | 4-hydroxybenzaldehyde | **4e** | 30 | 89 | 121 | 116–120 [28] |
| 6 | 4-dimethylaminobenzaldehyde | **4f** | 35 | 91 | 225–228 | 220–222 [28] |
| 7 | 3,4-dimethoxybenzaldehyde | **4g** | 30 | 92 | 180–183 | 179–181 [29] |
| 8 | 2-hydroxy 3-methoxybenzaldehyde | **4h** | 50 | 87 | 177–180 | 181–183 [29] |
| 9 | 3-nitrobenzaldehyde | **4i** | 25 | 92 | 226–228 | 229–231 [30] |
| 10 | 3-methoxybenzaldehyde | **4j** | 35 | 90 | 193–196 | 195–197 [31] |
| 11 | 2,4-dichlorobenzaldehyde | **4k** | 20 | 94 | 122 | 120–122 [29] |
| 12 | 4-methylbenzaldehyde | **4l** | 40 | 93 | 214–216 | 212–214 [31] |
| 13 | 4-chlorobenzaldehyde | **4m** | 25 | 95 | 179–181 | 173–175 [32] |
| 14 | 2-nitrobenzaldehyde | **4n** | 32 | 91 | 185–187 | 183–184 [33] |
| 15 | 3-nitrobenzaldehyde | **4o** | 35 | 93 | 179–182 | 183–185 [32] |
| 16 | 4-nitrobenzaldehyde | **4p** | 30 | 96 | 180–183 | 183–185 [32] |
| 17 | 4-hydroxybenzaldehyde | **4q** | 35 | 89 | 192–194 | 192–194 [34] |
| 18 | 4-dimethylaminobenzaldehyde | **4r** | 45 | 86 | 160–162 | 163–165 [35] |
| 19 | 2-chlorobenzaldehyde | **4s** | 45 | 83 | 194–197 | 193–194[33] |
| 20 | 3-chlorobenzaldehyde | **4t** | 50 | 84 | 180–182 | 177–178 [33] |
| 21 | 4-methoxybenzaldehyde | **4u** | 40 | 87 | 138–140 | 140–141 [30] |

**Scheme 2.** One-pot three-component reaction of malononitrile (**1**), aldehyde (**2**) and dimedone (**3a**) or ethyl acetoacetate (**3b**) catalysed by CuFe$_2$O$_4$@starch at room temperature.

**Scheme 3.** One-pot three-component reaction of 4-hydroxycoumarin (**8**), malononitrile (**1**) and aldehydes (**2**) catalysed by CuFe$_2$O$_4$@starch at room temperature.

**Table 3.** Three-component synthesis of different 3-amino-1-5-oxo-1,5-dihydropyrano[2,3-c] chromene-2-carbonitrile (**6a–l**) via condensation of malononitrile (**1**), various aldehydes (**2**) and 4-hydroxycoumarin (**5**) in the presence of CuFe$_2$O$_4$@starch at room temperature.

| entry | aldehyde | product | time (min) | isolated yields (%) | m.p. (obsd) (°C) | m.p. (lit) (°C) |
|---|---|---|---|---|---|---|
| 1 | 4-chlorobenzaldehyde | **6a** | 30 | 95 | 277–279 | 280–282 [36] |
| 2 | 4-nitrobenzaldehyde | **6b** | 32 | 91 | 253–255 | 256 [37] |
| 3 | 2-nitrobenzaldehyde | **6c** | 30 | 94 | 256 | 258–260 [36] |
| 4 | 3-nitrobenzaldehyde | **6d** | 37 | 89 | 232–234 | 237 [37] |
| 5 | 4-hydroxybenzaldhyde | **6e** | 40 | 80 | 268–271 | 272–274 [38] |
| 6 | 4-dimethylaminobenzaldehyde | **6f** | 45 | 82 | 222–223 | 220–222 [39] |
| 7 | 3,4-dimethoxybenzaldehyde | **6g** | 42 | 90 | 171–173 | 175–177 [38] |
| 8 | 4-methoxybenzaldehyde | **6h** | 50 | 92 | 238–240 | 234–236 [38] |
| 9 | 4-methylbenzaldehyde | **6i** | 45 | 87 | 233–235 | 231–233 [38] |
| 10 | 4-bromoenzaldehyde | **6j** | 30 | 89 | 234–236 | 230–234 [36] |
| 11 | 4-fluorobenzaldehyde | **6k** | 25 | 94 | 199–201 | 203 [37] |
| 12 | benzaldehyde | **6l** | 40 | 93 | 250–252 | 248 [37] |

friendliness, CuFe$_2$O$_4$@starch usage is fundamental in the synthesis of 4H-pyran. Savings in energy consumption, low cost, using ethanol as a green solvent, short reaction time, highly excellent yields and easy work-up due to using an external magnetic bar for detaching the bionanocatalyst from the reaction mixture are some of the advantages of this method. In addition, the simplicity of operation and the smaller amount of the chemical wastes make bionanocatalyst economically affordable for the synthesis of heterocyclic compounds. Due to the presence of CuFe$_2$O$_4$ nanoparticles in the bionanocatalyst, it has a high active surface area, which leads to increased catalytic activity.

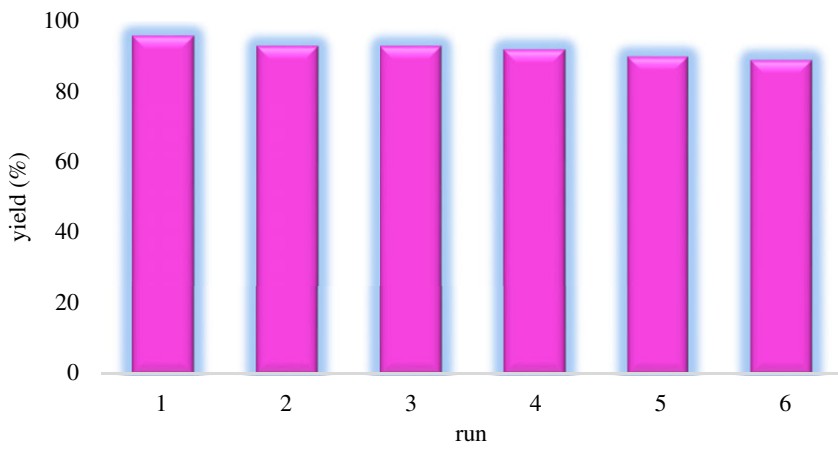

**Scheme 4.** One-pot three-component reaction of 2-hydroxy-1,4-naphthoquinone (**7**), malononitrile (**1**) and different aldehydes (**2**) catalysed by $CuFe_2O_4$@starch at room temperature.

**Figure 8.** Reusability of the $CuFe_2O_4$@starch bionanocomposite.

**Table 4.** Three-component synthesis of different 2-amino-5,10-dihydro-5,10-dioxo-*4H*-benzo[g]chromene-3-carbonitrile (**8a–j**) via condensation of malononitrile (**1**), various aldehydes (**2**) and 2-hydroxy-1,4-naphthoquinone (**7**) in the presence of $CuFe_2O_4$@starch at room temperature.

| entry | aldehyde | product | time (min) | isolated yields (%) | M.p. (Obsd) (°C) | M.P. (Lit) (°C) |
|---|---|---|---|---|---|---|
| 1 | 4-chlorobenzaldehyde | **8a** | 25 | 97 | 245–247 | 243–244 [11] |
| 2 | 2-chlorobenzaldehyde | **8b** | 30 | 90 | 240–242 | 238–240 [39] |
| 3 | 4-nitrobenzaldehyde | **8c** | 28 | 89 | 229–232 | 232–234 [11] |
| 4 | 2-nitrobenzaldehyde | **8d** | 33 | 87 | 246–248 | 242–244 [39] |
| 5 | 4-hydroxybenzaldhyde | **8e** | 37 | 89 | 263–265 | 258–260 [11] |
| 6 | 3,4-dimethoxybenzaldehyde | **8f** | 40 | 93 | 275–277 | 270–272 [39] |
| 7 | 4-methoxybenzaldehyde | **8g** | 38 | 87 | 260–263 | 257–259 [40] |
| 8 | 4-methylbenzaldehyde | **8h** | 45 | 88 | 259–262 | 254–257 [40] |
| 9 | 4-fluorobenzaldehyde | **8i** | 28 | 91 | 290–293 | 286–288 [39] |
| 10 | 4-bromoenzaldehyde | **8j** | 30 | 95 | 242–244 | 249–251 [11] |

Proceeding in three different reaction batches catalysed with $CuFe_2O_4$@starch in short reaction times and showing high efficiency are strong indicators of high performance and extensibility of the synthesized bionanocatalyst. The optimum reaction condition for those three different reactions catalysed by $CuFe_2O_4$@starch for synthesizing various derivatives of 4H-pyran is associated with the principles of green chemistry. This reaction takes place in ethanol as a solvent at room temperature without using energy, and the optimum amount of bionanocatalyst obtained in these reactions was 30 mg. Hence, at the end of the reaction, the easy purification process for the product and the possibility of re-using and recovering the catalyst without further decrease in catalytic activity is significant. Furthermore,

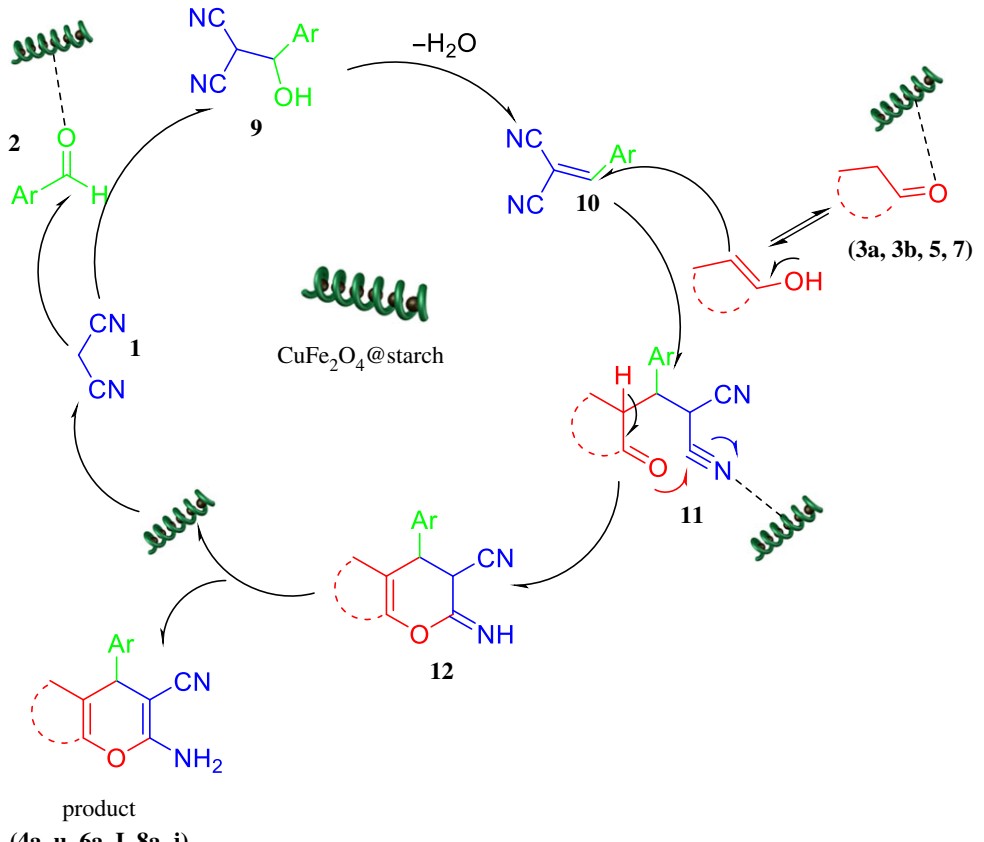

**Figure 9.** A plausible mechanism for the one-pot three-component reaction of different enolizable C-H activated acidic compounds (**3a, 3b, 5, 7**), malononitrile (**1**), aldehydes (**2**) catalysed by CuFe$_2$O$_4$@starch at room temperature.

using CuFe$_2$O$_4$@starch in terms of time and energy is valuable, which gives it the property of being scaled up in the pharmaceutical industry.

Data accessibility. The datasets supporting this article have been uploaded as part of the electronic supplementary material.

Authors' contributions. M.K. carried out the laboratory work, performed the synthesis experiments, prepared data analyses, participated in the design of the study and drafted the manuscript; M.B. and A.M. conceived of the study, designed the study, coordinated the study and helping in drafted the manuscript and interpretation of data. All authors gave final approval for publication.

Competing interests. We have no competing interests.

Funding. The authors received no specific funding for this work.

Acknowledgements. The authors gratefully acknowledge the partial support from the Research Council of the Imam Khomeini International University of Qazvin and the Iran University of Science and Technology.

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
