## [Reviewer comments · Royal Society Open Science]

Review History

RSOS-200385.R0 (Original submission)

Review form: Reviewer 1 (Tarek Abouelmaaty)

Is the manuscript scientifically sound in its present form?

Yes

Are the interpretations and conclusions justified by the results?

Yes

Is the language acceptable?

No

Do you have any ethical concerns with this paper?

No

Have you any concerns about statistical analyses in this paper?

Yes

Recommendation?

Major revision is needed (please make suggestions in comments)

Comments to the Author(s)

In this work, the authors reported on the synthesis of heterogeneous nanocatalyst based on a biopolymer followed by multicomponent reaction for the synthesis of 4H-pyran derivatives. The idea is interesting and off added value to researchers in the field of nano materials, however some flaws have been observed in the manuscript that i can outline as follow:

- English is poor and needs professional editing.
- Abstract should be modified to be more informative.
- Introduction failed to show the novelty of the current work in relation to previous publications.
- conclusion should be different than abstract and includes the optimum conditions for the obtained results.
- References are too many for an article, also ref 37 is not consistent with the style.

I recommend major revision

Review form: Reviewer 2

Is the manuscript scientifically sound in its present form?

Yes

Are the interpretations and conclusions justified by the results?

Yes

Is the language acceptable?

No

Do you have any ethical concerns with this paper?

No

Have you any concerns about statistical analyses in this paper?

No

Recommendation?

Accept with minor revision (please list in comments)

Comments to the Author(s)

In this manuscript authors have used starch as a biocompatible solid support to design magnetic nanobiocatalyst for three component coupling reaction for the synthesis of 4H-pyran derivatives. Utilization of bio support and magnetic separation are the advantages of this work. But several useful information are not provided in this manuscript. Thus, I recommend this manuscript for publication after a revision addressing the comments given below:

Introduction is not written properly. Several other catalysts which can catalyze this three component coupling reactions involving activated aromatic compounds, aromatic aldehyde and malononitrile should be referred. See and include: Dalton Trans., 2013, 42, 10515-10524; RSC Adv., 2012, 2, 11306-11317.

Electron microscopic image as presented in Figure 3 is of very low resolution. It is very hard to understand the particle size and shape from this image. Better quality image should be provided.

In the powder XRD pattern shown in Figure 4, peaks corresponding to the CuFe₂O₄ phase should be indexed.

Powder XRD pattern of the used CuFe₂O₄@starch catalyst after six reaction cycles should be provided.

Thorough English language editing is needed.

Decision letter (RSOS-200385.R0)

15-Apr-2020

Dear Professor Maleki:

Title: Green and efficient three-component synthesis of 4H-pyran catalyzed by CuFe₂O₄@starch as a magnetically recyclable bionanocatalyst
Manuscript ID: RSOS-200385

The editor assigned to your manuscript has now received comments from reviewers. We would like you to revise your paper in accordance with the referee and Subject Editor suggestions which can be found below (not including confidential reports to the Editor). Please note this decision does not guarantee eventual acceptance.

Please submit your revised paper before 08-May-2020. Please note that the revision deadline will expire at 00.00am on this date. If we do not hear from you within this time then it will be assumed that the paper has been withdrawn. In exceptional circumstances, extensions may be possible if agreed with the Editorial Office in advance. We do not allow multiple rounds of revision so we urge you to make every effort to fully address all of the comments at this stage. If deemed necessary by the Editors, your manuscript will be sent back to one or more of the original reviewers for assessment. If the original reviewers are not available we may invite new reviewers.

Royal Society of Chemistry
Thomas Graham House

Science Park, Milton Road
Cambridge, CB4 0WF
Royal Society Open Science - Chemistry Editorial Office

RSC Associate Editor:
Comments to the Author:
(There are no comments.)

RSC Subject Editor:
Comments to the Author:
(There are no comments.)

Reviewers' Comments to Author:
Reviewer: 1

Comments to the Author(s)

In this work, the authors reported on the synthesis of heterogeneous nanocatalyst based on a biopolymer followed by multicomponent reaction for the synthesis of 4H-pyran derivatives. The idea is interesting and off added value to researchers in the field of nano materials, however some flaws have been observed in the manuscript that i can outline as follow:

- English is poor and needs professional editing.
- Abstract should be modified to be more informative.
- Introduction failed to show the novelty of the current work in relation to previous publications.
- conclusion should be different than abstract and includes the optimum conditions for the obtained results.
- References are too many for an article, also ref 37 is not consistent with the style.

I recommend major revision

Reviewer: 2

Comments to the Author(s)

In this manuscript authors have used starch as a biocompatible solid support to design magnetic nanobiocatalyst for three component coupling reaction for the synthesis of 4H-pyran derivatives. Utilization of bio support and magnetic separation are the advantages of this work. But several useful information are not provided in this manuscript. Thus, I recommend this manuscript for publication after a revision addressing the comments given below:

Introduction is not written properly. Several other catalysts which can catalyze this three component coupling reactions involving activated aromatic compounds, aromatic aldehyde and malononitrile should be referred. See and include: Dalton Trans., 2013, 42, 10515-10524; RSC Adv., 2012, 2, 11306-11317.

Electron microscopic image as presented in Figure 3 is of very low resolution. It is very hard to understand the particle size and shape from this image. Better quality image should be provided.

In the powder XRD pattern shown in Figure 4, peaks corresponding to the CuFe₂O₄ phase should be indexed.

Powder XRD pattern of the used CuFe₂O₄@starch catalyst after six reaction cycles should be provided.

Thorough English language editing is needed.

Author's Response to Decision Letter for (RSOS-200385.R0)

See Appendix A.

Decision letter (RSOS-200385.R1)

Dear Professor Maleki:

Title: Green and efficient three-component synthesis of 4H-pyran catalyzed by CuFe₂O₄@starch as a magnetically recyclable bionanocatalyst
Manuscript ID: RSOS-200385.R1

It is a pleasure to accept your manuscript in its current form for publication in Royal Society Open Science. The chemistry content of Royal Society Open Science is published in collaboration with the Royal Society of Chemistry.

RSC Associate Editor
Comments to the Author:
(There are no comments.)

Reviewer(s)' Comments to Author:

Appendix A

Dear Dr. Laura Smith, Publishing Editor:

Thank you so much for your kind e-mail message on April 15, 2020; and very useful Reviewers' comments and editorial suggestions on our manuscript (ID: RSOS-200385). It is our great pleasure to submit the enclosed revised manuscript to be considered for publication in *the Royal Society Open Science*. We have modified the manuscript accordingly (as highlighted by the yellow color in Text), and detailed corrections are listed point by point as follows:

Comments from the editors and reviewers:

-Reviewer 1

In this work, the authors reported on the synthesis of heterogeneous nanocatalyst based on a biopolymer followed by multicomponent reaction for the synthesis of 4H-pyran derivatives. The idea is interesting and off added value to researchers in the field of nano materials, however some flaws have been observed in the manuscript that I can outline as follow:

- English is poor and needs professional editing.

Answer: We greatly appreciate your careful reading and reviewing our manuscript. The whole Text was edited by an official English translator.

- Abstract should be modified to be more informative.

Answer: Thank you for your comment. It was done in the main manuscript in the abstract section.

- Introduction failed to show the novelty of the current work in relation to previous publications.

Answer: Thank you for your insightful comment. It was done in the main manuscript in the introduction section.

- Conclusion should be different than abstract and includes the optimum conditions for the obtained results.

Answer: Thank you for your comment. It was done in the main manuscript in the conclusion section.

- References are too many for an article, also ref 37 is not consistent with the style.

Answer: Thank you for your comment. The number of references was decreased as much as possible.

Reviewer 2

In this manuscript authors have used starch as a biocompatible solid support to design magnetic nanobiocatalyst for three component coupling reaction for the synthesis of 4H-pyran derivatives. Utilization of bio support and magnetic separation are the advantages of this work. But several useful information are not provided in this manuscript. Thus, I recommend this manuscript for publication after a revision addressing the comments given below:

- Introduction is not written properly. Several other catalysts which can catalyze this three component coupling reactions involving activated aromatic compounds, aromatic aldehyde and malononitrile should be referred. See and include: Dalton Trans., 2013, 42, 10515-10524; RSC Adv., 2012, 2, 11306-11317.

Answer: Thank you for your comment. Sure, they are worthy reports and were used in the introduction section.

- Electron microscopic image as presented in Figure 3 is of very low resolution. It is very hard to understand the particle size and shape from this image. Better quality image should be provided.

Answer: Thank you for your comment. The Electron microscopic image of bionanocatalyst was replaced (Figure 3).

- In the powder XRD pattern shown in Figure 4, peaks corresponding to the CuFe_2O_4 phase should be indexed.

Answer: Thanks for your admirable attention. The indices of the peaks were shown in XRD patterns (Figure 4).

- Powder XRD pattern of the used CuFe_2O_4 @starch catalyst after six reaction cycles should be provided.

Answer: Thank you for your comment. The powder XRD pattern of the reused catalyst was added in the main manuscript (Figure 4).

- Thorough English language editing is needed.

Answer: Thank you for your comment. The whole Text was edited by an official English translator.